# The Case (or Not) for Life in the Venusian Clouds

**DOI:** 10.3390/life11030255

**Published:** 2021-03-20

**Authors:** Dirk Schulze-Makuch

**Affiliations:** 1Astrobiology Research Group, Center for Astronomy and Astrophysics (ZAA), Technische Universität Berlin, Hardenbergstr. 36, 10623 Berlin, Germany; schulze-makuch@tu-berlin.de; Tel.: +49-30-314-23736; 2German Research Centre for Geosciences (GFZ), Section Geomicrobiology, 14473 Potsdam, Germany; 3Department of Experimental Limnology, Leibniz-Institute of Freshwater Ecology and Inland Fisheries, (IGB), 12587 Stechlin, Germany; 4School of the Environment, Washington State University, Pullman, WA 99163, USA

**Keywords:** Venus, life, atmosphere, clouds, extreme environment

## Abstract

The possible detection of the biomarker of phosphine as reported by Greaves et al. in the Venusian atmosphere stirred much excitement in the astrobiology community. While many in the community are adamant that the environmental conditions in the Venusian atmosphere are too extreme for life to exist, others point to the claimed detection of a convincing biomarker, the conjecture that early Venus was doubtlessly habitable, and any Venusian life might have adapted by natural selection to the harsh conditions in the Venusian clouds after the surface became uninhabitable. Here, I first briefly characterize the environmental conditions in the lower Venusian atmosphere and outline what challenges a biosphere would face to thrive there, and how some of these obstacles for life could possibly have been overcome. Then, I discuss the significance of the possible detection of phosphine and what it means (and does not mean) and provide an assessment on whether life may exist in the temperate cloud layer of the Venusian atmosphere or not.

## 1. Introduction

Ever since Morowitz and Sagan [1] suggested that life may exist in the lower Venusian atmosphere, many authors looked into that possibility [2,3,4,5,6,7,8,9,10,11] and made incremental progress understanding the environmental conditions in the Venusian atmosphere, and the possibility of it being inhabited by microbial life. The presence of an aerial biosphere may seem strange to an observer from Earth, because on Earth, the atmosphere appears to serve only as a temporary habitat [12], as reproduction has not been demonstrated in Earth’s atmosphere [13]. This assessment is further supported by a study from Amato et al. [14], which indicated the lack of evidence for bacterial cell division in the atmosphere through meta-transcriptomics analysis. However, an earlier field study seemed to indicate that bacteria can grow and reproduce in cloud droplets [15] and earlier laboratory studies indicated that limited cell divisions of the facultative anaerobic bacterium *Serratia marcescens* occurred in an airborne state [16,17]. It would not be surprising if Earth’s atmosphere is not utilized as a permanent habitat, because the environmental conditions on Earth’s surface are well-suited for life and any microorganisms in the atmosphere would be deposited back on the surface by precipitation, usually within a few days [18]. The situation is different on Venus.

Whether there is or was life on Venus is a highly speculative assertion, simply because we do not have much data from Venus and its atmosphere despite many missions to this planet, especially during the Soviet era. Our understanding of Venus as an astrobiological target did incrementally increase during the last decades, including a constant trickle of papers, e.g., References [1,3,4,5,6,7,8,9,10,11], but only with the claimed detection of phosphine in the Venusian atmosphere by Greaves et al. [2], pointing to the possibility of life, did that question gain much attention, both in the scientific community and public discourse. A flurry of papers has been produced since that announcement, so it seems timely to take a first stock of where we are standing and what should be the next steps for Venusian research, about 7 months after the initial announcement.

## 2. The Environmental Conditions in the Venusian Clouds

In early Solar System history, Venus was in its habitable zone, and oceans likely existed on the Venusian surface [19], as they did on Earth and likely Mars [20]. In fact, if life did not originate on Venus independently under roughly similar conditions as on Earth at that time, it could have been transported from either Earth or Mars [19]. It is unknown how long the Venusian surface remained habitable, while the Sun’s luminosity increased, but more recent studies [9,21] suggest that Venusian oceans may have possibly lasted until as recently as 715 million years ago. It has been speculated that microbial life could have adapted to the extreme environmental conditions in the Venusian atmosphere by natural selection as the last possible (borderline) habitat remaining on the planet [6]. Researchers focused on the lower cloud deck as a potential microbial habitat where temperatures range between 40 and 90 °C at a pressure of roughly 1 bar [22], consistent with a habitat for thermophilic microbes on Earth. The cloud decks of Venus are also much larger in extent, providing more continuous and stable environments than clouds on Earth [7] and contain particles (aerosols) that are projected to last for several months in the Venusian atmosphere [3]. The Venusian atmosphere is in slight thermodynamic disequilibrium as oxygenated chemical species, such as SO_2_ and O_2_, coexist with reducing species, such as H_2_S—and possibly also PH_3_ based on the latest possible detection [2]. However, these chemical compounds are very heterogeneously distributed in the Venusian atmosphere [23,24,25]. In addition, methane has been detected at a concentration of 980 ppm at an altitude of about 50 km by the Pioneer-Venus probe [26]. That detection has been questioned [26], however, and no plausible explanation for its presence has been advanced [23], but if it is real, it would be a potential additional atmospheric biosignature.

Not all environmental parameters in the lower Venusian atmosphere are favorable to life, however. Quite the contrary. One critical parameter is water activity, which can be used as a measure of available water from a microbial perspective. Water activities range between 0 and 1, with 1 being the water activity of pure water. Salt water has a water activity slightly lower than 1, because some of the water molecules are inaccessible for microbes due to the dissolved salts within the water. Honey typically has a water activity too low for microbial consumption. Therefore, it does not spoil, even though it is highly nutritious. In regard to atmospheres, water activity is equal to the equilibrium relative humidity divided by 100 and is usually measured in the field as such. Determining the water activity in the Venusian atmosphere, where no direct measurements can be taken, is obviously much more challenging. Estimates of the concentration of sulfuric acid in the Venusian clouds range between 75% and 96% [27], which should reduce water activities to values below 0.1, clearly out of the range for the three domains of life on Earth [28]. The water activity limit of life on Earth seems to be about 0.605 [29], which is indisputably a high, perhaps impossible, barrier to reach within the Venusian atmosphere. Two studies may shed some additional insights. In one study, lichen grew under Martian radiation-sheltered niche conditions, despite water activity being most of the time below the known threshold for life on Earth [30], meaning that it is sufficient if the threshold is overcome during a fraction of exposure time. In another study, the water activity in a liquid asphalt lake was measured to be 0.49, yet life thrived in it [31]. While the overall measured water activity was below the threshold, a whole ecosystem of bacteria was discovered in microdroplets of water within the oil matrix [32]—the point being that we have to look at the correct scale and search for possible microenvironments.

Nevertheless, extrapolated water activities in the Venusian atmosphere are so low that it is hard to believe that any Earth organism would come even close to mastering them. It might be that we have to invoke a different biochemistry to overcome the constraint imposed by water activity. Limaye et al. [33] pointed out that the abundance of sulfuric acid at the Venusian aerosols, which were proposed to harbor life, is based on indirect measurements and computer simulations, and recent insights seem to indicate the presence of chemical compounds other than sulfuric acid. After all, at the locality and scale where it counts, sulfuric acid concentrations may be lower and water activities higher than is usually assumed.

Still, it is difficult to see how a viable range, from an Earth biology perspective, could be reached. In addition, the hyperacidity of the cloud environment with sulfuric acid as an extremely aggressive chemical compound, makes possible life under these conditions a hypothesis hard to defend. Seager et al. [11], for example, pointed out that crucial biochemical compounds essential to life on Earth are unstable in sulfuric acid, which includes the major building blocks of life such as carbohydrates, nucleic acids, and proteins. Are we back to the notion that we have to invoke a different biochemistry and that life as we know it simply cannot meet the challenge?

Clearly, these are hurdles not easy to overcome. Nevertheless, several authors tried to come up with biological solutions to the challenges for a putative Venusian biosphere. For example, Schulze-Makuch et al. [6] suggested that putative microbial cells might be protected by elemental sulfur, thus being insulated from the sulfuric acid environment (elemental sulfur also being critical for photosynthesis, see below). Petrova [34] added that elemental sulfur is not wetted by sulfuric acid, and thus could provide protection from the concentrated sulfuric acid environment with cells only adhering to sulfuric acid droplets rather than being entirely enveloped [11]. Nutrient requirements could also be a problem for putative microbial life in the Venusian atmosphere, especially trace metals (Figure 1). However, it seems that phosphorus may be common, perhaps as common as sulfur in the Venusian atmosphere, and there are examples on Earth of microbes that obtain all their carbon and nitrogen needs from the atmosphere [11]. X-ray fluorescence measurements by the Venera 13 and 14, and Vega 1 and 2 descent probes not only found sulfur, but also phosphorus, chlorine, and iron—with up to as much phosphorus as sulfur in the lower clouds below 52 km [27], most likely being in the form of phosphoric acid (H_3_PO_4_) [35]. Cockell [4] pointed out that, in terms of elemental requirements for life (e.g., C, N, P), the lower clouds of Venus are attractive sites for biology, though available hydrogen may be a problem.

Clearly, the suggestion that the lower cloud deck is a habitat, and one that is hosting life, is a difficult one to maintain, especially if we base our analyses on life as we know it and the biochemical adaptations we are familiar with. Furthermore, microbes do not live as individuals, but are part of a larger biosphere, which increases our challenges even more, because we would have to propose a biosphere that can thrive in the Venusian clouds despite their extreme environmental conditions. Nevertheless, hypotheses have been put forward.

## 3. Proposed Adaptations of Microbial Life to the Venusian Cloud Environment

In the last decades of analog research on Earth, we learned much on how microbes are able to use their environmental resources to an amazing degree, circumnavigate challenges, or even co-opt in principle detrimental factors to their advantage [36,37]. One of these potentially detrimental factors is UV irradiation, particularly in an atmospheric environment. However, for putative life on Venus, UV irradiation may actually be an asset. The reason is that the Venusian cloud layer contains large amounts of elemental sulfur, particularly cycloocta sulfur (S_8_) [38,39] (see also Figure 2 in Reference [6]), which has the intriguing property to adsorb UV irradiation and re-radiate it in the visible light spectrum.

The aerosols, also called mode 3 particles, which are present in the lower cloud deck of Venus seem to be coated to a large degree by elemental sulfur based on spectral analyses [40]. It has been proposed that these aerosols could be microbes that use elemental sulfur to power anaerobic photosynthesis reactions ([6], Equation (1)), a highly energetic pathway that could be able to sustain a permanent microbial biosphere in the Venusian clouds. The authors further elaborated that the sulfur that is oxidized during photosynthesis might later be reduced by chemoautrophic microorganisms to close the nutrient cycle. Limaye et al. [10] supported that notion by reporting that more than half of the UV irradiation that the Venusian atmosphere receives is absorbed by an unknown mechanism and speculated that this phenomenon could be the result of an energy capture process by an aerial biosphere. The reason is that none of the abiotic explanations of the UV absorber (including S_8_) can fully explain its abundance and spectrum. The possibility of anaerobic photosynthesis as a main energy capture process would also be supported by the super-rotation of the Venusian atmosphere, which cuts the nighttime significantly, allowing shorter periods between light and dark [41].
2 H_2_S + CO_2_ + light → CH_2_O + H_2_O + S_2_(1)

Seager et al. [11] added and elaborated on the advanced hypothesis by Schulze-Makuch et al. [6], pointing out that the coating of the cells would also have to include hydrophilic filaments in addition to the elemental sulfur to allow the putative microorganism to uptake critical liquids. They further suggested a life cycle to address the problem of microbial cells falling through the clouds towards the surface, where they would eventually die and be permanently removed from the atmosphere. In their conceptual model, the microbial cells would dry out during settling and become desiccated spores, which later would be returned to the cloud layer by gravity waves. Once back in the cloud layer, they would rehydrate by cloud condensation to complete the cycle.

That model would have the advantage that the loss rate of microbial cells would be much lower compared to the hypothesized microbial ecosystem by Schulze-Makuch et al. [6], in which the reproduction rate would have to compensate for the rate at which the aerosols drop out of the cloud layer. The constraints on the required reproduction rate to maintain a constant microbial population in the Venusian clouds would be less limited in the Seager et al. model [11], but it would require that the proposed microorganisms can form spores to counter the extreme environmental conditions when sinking into the lower haze layer of the Venusian atmosphere.

## 4. The Claimed Detection of Phosphine

Phosphine (PH_3_) was claimed to be detected in the Venusian atmosphere at a concentration of about 20 parts per billion, using thorough spectral analysis and observations from two different Earth-based telescopes in 2017 and 2019 [2]. More recently, the detection has been corrected down to peak concentrations of 5–10 ppb and a global concentration of 1–4 ppb by the authors from the original paper [42]. However, concerns about the spectral analysis [43], in particular that the PH_3_ detection could be misidentified sulfur dioxide, have been raised [44,45,46], mostly because the absorption lines of PH_3_ (266.94 GHz) and SO_2_ (~267.5 GHz) are close. The original authors continued to insist that their analytical methods are correct, and that the phosphine detection is valid [47]. SO_2_ is the third most abundant gas in the lower atmosphere of Venus and usually occurs at least in the ppm range, but the Greaves et al. [2] study claimed it was below their detection threshold and interpreted the identified spectral peak to be PH_3_. Rimmer et al. [48] suggested that the lack of SO_2_ is due to its dissolution in the clouds because of the presence of hydroxide salts. However, this does not necessarily verify the PH_3_ detection, and no resolution of the scientific controversy about the validity of the detection is to be expected any time soon. In principle, more than enough phosphorus should be present (in oxidized form) within the Venusian atmosphere to explain the reported detection of PH_3_ [49]. From a viewpoint of P as purely a nutritional requirement, enough phosphorus should be present to allow the presence of microbial life [50], but certainly P availability is only one of many constraints for life.

Sousa-Silva et al. [51] argued for PH_3_ being a suitable biosignature for an oxidized atmosphere such as Venus, because of the apparent lack of “abiotic” false positives. They also claimed that phosphine has uniquely identifiable spectral features in the infrared range. However, this is not the spectral range Greaves et al. [2] used to detect phosphine. As the phosphine detection has been marred in controversy [42,43,44,45,46,47,48], an independent verification of the claimed PH_3_ detection is warranted. Especially insightful would be if a different methodology could be employed, such as trying to detect phosphine in the infrared range of the spectrum. An independent confirmation could also derive from a re-analysis of the Large Probe Neutral Mass Spectrometer (LNMS) on board of the Pioneer-Venus mission, as done by Mogul et al. [52]. In their recently published paper, they claimed that their analysis is suggestive of PH_3_ and H_2_S being present in the middle clouds, based on a peak fitting model that uses data points within the LNMS dataset to estimate the full-width half-maximum and peak heights of chemical reference and target species. Their re-analysis of the original data also suggested the presence of other reduced chemical compounds such as carbon monoxide, ethane, and nitrogen species in various oxidation states, thus pointing to more complex redox disequilibria than only from the presence of PH_3_ and H_2_S.

Phosphine is a colorless gas, which is toxic to aerobic organisms including humans, and on Earth it is associated with anaerobic life [53,54,55]. Exact processes involved are unclear, but Bains et al. [56] suggested that phosphate-reducing bacteria could obtain energy from the reduction of phosphate (HPO_4_^2−^) to phosphite (HPO_3_^2−^) by coupling phosphate-reduction to NADH oxidation. The phosphine would then be produced by the combined action of phosphate reducing and phosphite disproportionating bacteria.

The claimed detection of phosphine in the oxidizing Venusian atmosphere is surprising, because, if real, it would mean that there has to be a pathway that consistently produces phosphine given its reactivity with other chemical compounds and its vulnerability of being broken apart by UV irradiation. Greaves et al. [2] and Bains et al. [57] considered a total of 74 potential abiotic phosphine production pathways, including gas- and cloud-phase reactions, reactions with sulfur haze, and reduction of phosphate minerals at the surface of the planet or as dust in the atmosphere. Their emphasis was on possible gas- and cloud-phase reactions, which included reduction reactions of H_3_PO_4_ to PH_3_, reduction reactions of P_4_O_10_ to PH_3_, reduction reactions of P_4_O_6_ to PH_3_, disproportionation of H_3_PO_3_ (in cloud droplets) and P_4_O_6_ in the gas phase, and reduction reactions of H_3_PO_3_ in the droplets to PH_3_.

Bains et al. [57] also looked at potential photochemical pathways, lightning as a potential source, subterranean sources such as volcanic outgassing, and asteroid or cometary impacts, but none explained the amount of detected phosphine. After completion of their investigation, they concluded that their analyses left either (A) an unknown geochemical or photochemical pathway or (B) biology in the clouds of Venus as the only explanations. Cockell et al. [58] reiterated that the low water activity makes the Venusian clouds uninhabitable to known life, while Izenberg et al. [59] concluded that life on Venus cannot be excluded as an option. They came up with a non-zero likelihood, to be exact between 10^−8^ and 10%, depending on the underlying assumptions.

Regarding Option A, it seems fair to state that potential abiotic geochemical or photochemical pathways are likely to exist, given the rich chemical endowment of Venus and the possibility of many heterogeneous reactions. For example, Catling [60] suggested that PH_3_ might be made via pathways from phosphorus trioxide (P_4_O_6_), which may go through a phosphorous acid (H_3_PO_3_) intermediary. Specifically, he suggested that the following reaction pathways may be involved: P_4_O_6_ + 6H_2_O → PH_3_ + 3H_3_PO_4_(2)
or, similarly, with hydrochloric acid instead of water:P_4_O_6_ + 6HCl → 2 H_3_PO_3_ + 2PCl_3_(3)

From the reaction described in Equation (3), phosphorous acid (H_3_PO_3_) can decompose into phosphoric acid (H_3_PO_4_) and phosphine (PH_3_). Bains [61] suggested that there would be too little water at the prevailing temperatures in the atmosphere, given an estimated relative humidity of 0.015%, for reaction (2) to be favorable. It would require about 200 kJ/mol under Venus atmospheric conditions. The feasibility of Equation (3) is difficult to assess, because the PCl_3_ concentration in the Venusian atmosphere is unknown. It has to be realized that an amount of 1–4 ppb for phosphine is relatively low, so an abiotic pathway seems certainly possible, or even likely. In that context, we also have to realize that Venus is largely still an alien planet, and we do not know much about the chemistry in its lower atmosphere. The majority of the missions to Venus took place during the Soviet era (1960–1980s), and there is a lot of uncertainty regarding both atmospheric gas compositions and abundances.

Regarding Option B, we have to acknowledge that life in the extreme environmental conditions of the lower Venusian atmosphere seems to be extremely challenging. As pointed out above, the hyperacidity is much stronger than in any natural environment on Earth, and no organism on Earth is known to thrive at water activities estimated to exist in the Venusian clouds. No organisms on Earth could withstand the acidity of an even 75% solution of sulfuric acid, which would reduce organic carbon compounds without any protection into elemental carbon. Thus, in order for life to thrive at these conditions, either some adaptation has to be proposed that has no analog in Earth’s biology or a different biochemistry has to be involved.

The latter possibility seems unlikely as life probably originated under similar conditions on Earth and Venus, or it might have been transported from Earth to Venus early in Solar System history. Microbial life might have existed in early Venusian oceans, but whether life could adapt to the current conditions is unclear. It depends to some degree on the natural history of the planet and how fast the runaway greenhouse effect occurred on our neighboring planet. If it was triggered by a cataclysmic event such as a huge asteroid impact or sequence of several of such impacts (the retrograde rotation of Venus may still be a consequence of such an encounter [62]), or by a global volcanic trigger [21], then it would he highly doubtful that an early biosphere could have survived. However, if there was enough time for natural selection to come up with better and better solutions to cloud-based life and increasingly acidic conditions, then this may be a possibility.

There are only a few hyper-acidic locations on Earth, such as the Dallol geothermal area, and even this area is not as hyper-acidic and is much wetter than the Venusian atmosphere [63], so there would be no reason for microbial life on our planet to evolve the needed capability. But could life adapt to these hyper-acidic conditions? It would have to be a multi-extremophilic microbe. Enough energy should be available through photosynthesis to come up with potential energy-requiring biochemical pathways, but we do not know how they would work. However, that does not mean they cannot exist.

There is one other intriguing detail about the new finding by Greaves et al. [2] that deserves mentioning. The phosphine distribution was heterogeneous, it was detected near the temperate latitudes, but not in the polar area. As the authors point out, this would be consistent with a biological explanation (but certainly no proof), because the atmospheric circulation patterns in the mid-latitudes would offer the most stable environment for life.

However, assuming the phosphine detection was real, the main problem with the abiotic explanation (and perhaps in favor of a biological explanation) is that PH_3_ is extremely difficult to produce in the oxidizing Venusian atmosphere because the strongest natural reducing agent is H_2_. Biology, however, can use agents that are more reducing, such as iron-sulfur proteins [57]. An iron-sulfur redox metabolism in the clouds of Venus was also suggested by Limaye et al. [10]. Of course, that requires a significant amount of extra energy, but again, photosynthesis as a metabolic pathway could easily provide the needed energy.

## 5. Discussion and Next Steps

So where does this leave us? The claimed presence of phosphine in the oxidizing atmosphere of Venus is just astounding, especially because the gas has not been detected previously on any other terrestrial planet besides Earth. However, that does not prove the presence of biology. There are many unknowns about our “twin planet”, which remains largely alien to us. Many processes and chemical reactions that are likely occurring in the Venusian atmosphere and also on the Venusian surface are not well understood. There are many puzzles and open questions, no matter whether we adhere to an unknown chemical pathway or biology as possible solutions to the observations made. For example, if these observations are due to abiotic chemistry, how could PH_3_ be continuously produced in the oxidizing atmosphere of Venus? If they are due to biology, this is: how could life permanently cope not only with the difficulties of being airborne, but also with hyperacidity, extreme lack of water, and a possible lack of critical nutrients? These challenges to life are very high, and thus Cockell et al. [64] argued that there is no good reason to entertain the biological hypotheses. They advised that the original authors should not have evoked it, but by doing so, Greaves et al. [2] caused a lot of media hype. However, as pointed out above, Venus likely had oceans on its surface in the past [9] and if life did not originate on Venus independently, it could have been transported from Earth into another habitable environment on Venus [13]. Therefore, I consider it reasonable that life existed on Venus at some point in the distant past, the more difficult hypothesis to defend is whether it could have adapted to the currently existing environmental conditions in the lower atmosphere. In my view, we have an anomaly in the sense of Cleland [65], that in principle could be caused by biology, even if seemingly unlikely, and we need to further investigate this “anomaly”.

The first step of the ensuing investigation should be to independently verify the detection of PH_3_. This is critical, because otherwise the possibility that SO_2_ was misidentified as PH_3_ looms large. Aside from trying to detect phosphine in the infrared range and by confirming it with LNMS mass spectra, I would suggest searching for diphosphine (P_2_H_4_) in the Venusian atmosphere. P_2_H_4_ is an intermediate in the photolysis reaction of phosphine to phosphorus and hydrogen [66], and thus should be present in the Venusian atmosphere if PH_3_ is present. Thus, efforts should be undertaken to also search for P_2_H_4_ in Venusian spectra.

We also have to think about possible mechanisms of how life could thrive in the extremely challenging environmental conditions in the Venusian clouds. The efforts by Schulze-Makuch et al. [6], Limaye et al. [10], and Seager et al. [11] are first steps toward this goal. Life under hyper-arid conditions on Earth, analogous to Martian environments, has resulted in amazing evolutionary adaptations, such as relying on deliquescence as a sole source of water for microbes [67]. What mechanisms could be envisioned in the Venusian clouds, especially on how to adapt to hyperacidity and the extreme lack of liquid water? Experiments are conducted to assert how far microorganisms can adapt to Martian environmental conditions, particularly high perchlorate concentrations [68], to assess the feasibility of life on Mars. Analogously, experiments should be conducted on how far selected acidophilic organisms can adapt from one generation to the next to higher and higher sulfuric acid concentrations. Also, it would be interesting to find out how microorganisms adapt to a lack of trace metals required for life as we know it. What are the microbial needs and are there ways to compensate for a lack of say magnesium, manganese, and even molybdenum? Are there biochemical ways we have not considered yet how microbes might cope to low water-activity environments? Can we at least theoretically envision how microbial life, despite the obvious challenges, could get its critical water in the Venusian atmosphere? Our results could be that we have to invoke not only biochemical pathways unknown from life as we know it, but an entirely new biochemistry. To do so would become highly speculative, yet we also have to recognize our limitation of knowing only one type of life (ours), even if it is incredibly diverse. But even from this limited dataset, it is obvious that life is amazingly adaptive to environmental changes. It will be one of the most exciting scientific endeavors to find out what these limitations of life are, no matter whether life exists at Venus or not.

In the last decades, there have not been many missions to Venus as astrobiologists and planetary scientists were focused on Mars and icy moons in the outer Solar System when searching for extraterrestrial life. Our next-door neighbor does, however, represent an intriguing target, especially since Venus used to be located in the habitable zone around our Sun, and may have had oceans and possibly also living organisms on its surface, perhaps for a very long time. The last possible outpost of life on Venus can only be in the temperate cloud decks of the lower Venusian atmosphere—given the even more extreme conditions on the surface and in the subsurface of that planet—and if it is there, it has to be in the form of aerosol particles floating in the more benign cloud layer (which is still incredibly extreme). A mission is overdue to analyze the interior of these aerosols to find out about their composition (after an earlier attempt failed with NASA’s Pioneer-Venus mission when the pyrolyzer jammed). If those aerosols contain organic compounds, a sample return mission should be launched to bring these particles back for further analysis on Earth. This can be done with a Stardust-type mission as suggested earlier [5], or with some other mission architecture that involves, for example, balloons, tethers [69], or aerial platforms to do the sample collection [70,71]. Whether Venus holds life or not, it is an intriguing planet to investigate.

## Figures and Tables

**Figure 1 life-11-00255-f001:**
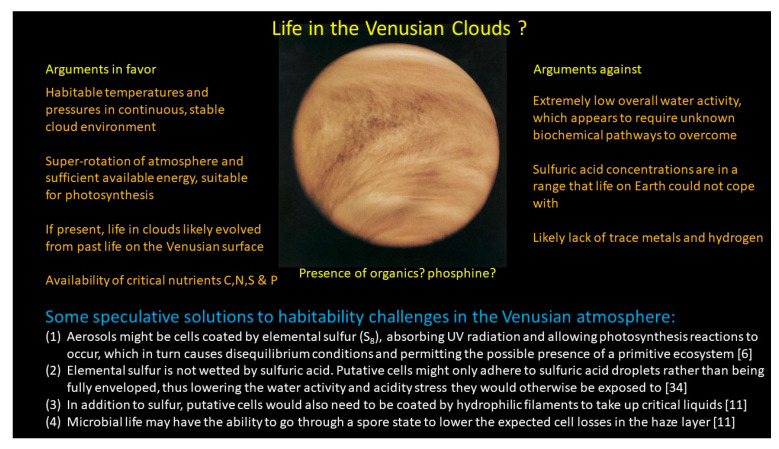
Major arguments for and against life on Venus, and some scientific speculations as to how some of the habitability challenges could be addressed by putative life in the Venusian cloud layer. In addition to resolving the question of the presence of organic compounds and phosphine, there are many other open questions regarding the Venusian environment, which would need to be resolved. Image shows cloud structure and presence of UV absorber (black streaks) as revealed by the Venus-Pioneer probe in 1979 (Credit: NASA).

## Data Availability

Date is contained within the article.

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
