# Peer review of "The Case (or Not) for Life in the Venusian Clouds"

_life, 2021, doi:10.3390/life11030255_

Round 1
Reviewer 1 Report
Comments to the author attached in a separate docx file.

Author Response
Response to Reviewer 1
The article titled “The Case (or not) for Life in the Venusian Clouds” summarizes and provides a commentary on the very interesting discovery of phosphine (PH3) in the clouds of Venus and the habitability of the planet in general.
The commentary makes reasonable assumptions and arrives at compelling conclusions, both of which are supported by (mostly) appropriate referencing.
Thank you
Specific points:
Line 30: Regarding the cell division in the clouds (in situ). I would not dismiss the possibility of the in-situ cell division inside the cloud droplets.
The author is correct that some microbes (both inside and outside water cloud droplets on Earth) are found to be metabolically active, even though there is no direct evidence, as of yet, of cell division in situ in the clouds. Nevertheless the possibility for active cell division in situ the clouds remains. The only work on the possibility of the active cell division in aerosols, that the reviewer is aware of, come from early laboratory studies that provide evidence for active reproduction of microbial cells (Serratia marcescens) in aerosols few micrometres in diameter (ref1, ref2). The cells divided 2-3 times in the 6 um aerosols. Propagations beyond three generations were not observed, possibly due to nutrient limitations or accumulation of metabolic waste products within the aerosol droplets (ref2).
ref1. Dimmick, R. L., Wolochow, H., & Chatigny, M. A. (1979). Evidence that bacteria can form new cells in airborne particles. Applied and Environmental Microbiology, 37(5), 924-927.
ref2. Dimmick, R. L., Wolochow, H., & Chatigny, M. A. (1979). Evidence for more than one division of bacteria within airborne particles. Applied and environmental microbiology, 38(4), 642-643.
Wording has been added and the two references are included now and mentioned in the text, as well as the work by Sattler et al. (2001)
Line 74: There are several other good habitats that the author could consider in his discussion. Sulfate reducing bacteria were found living (and presumably reproducing) in a (close to) saturated solution of MgCl2 where water activity is 0.4 (previous record for active cell division and metabolism is ~0.6) (ref3).
ref3. Steinle, Lea, et al. "Life on the edge: active microbial communities in the Kryos MgCl 2-brine basin at very low water activity." The ISME journal 12.6 (2018): 1414-1426.
That study is intriguing, but I wonder whether the observations are due to a mixing effect and whether the microorganisms are truly active and reproducing. The work of my own research group shows how difficult it is for life to cope with chaotropic solutions, including MgCl2 solutions, and we are nowhere getting close to saturation with the most adaptive organism, a yeast. See for example:
- Waajen, A.C., Heinz, J., Airo, A., and Schulze-Makuch, D. (2020) Physicochemical salt solution parameters limit the survival of Planococcus halocryophilus in Martian cryobrines. Frontiers Microbiology 11, #1284, doi: 10.3389/fmicb.2020.01284.
- Heinz, J., Krahn, T., and Schulze-Makuch, D. (2020) A new record for microbial perchlorate tolerance: fungal growth in NaClO4 Brines and its implications for putative life on Mars. Life 10 (5): 53, https://doi.org/10.3390/life10050053
- Heinz, J., Waajen, A.C., Airo, A., Alibrandi, A., Schirmack, J., and Schulze-Makuch, D. (2019) Bacterial growth in chloride and perchlorate brines: Halotolerances and salt stress responses of Planococcus halocryophilus. Astrobiology 19 (11): 1377-1387, doi.org/10.1089/ast.2019.2069
So, decided to leave the point as is with the two examples given
The estimated water activity in Don Juan Pond on Antarctica is likely below 0.45 aw but could be between 0.28 aw (+25C) - 0.61 aw (–50C) if we take under consideration the dominant salt there (CaCl2). As far as the reviewer knows, it is unclear if Don Juan Pond is inhabited.
It is generally assumed that there is no active life in the Don Juan Pond, but only microbes that fall in from the surroundings (and die). But it would be very interesting to investigate further and I tried to get samples since years….
Lines 88-89: “(…) is based on indirect measurements and computer simulations and recent insights seem to indicate the presence of chemicals compounds other than sulfuric acid." – this is very interesting statement that warrants the citations – citations should be provided for those recent studies.
The citation for that statement is given (Limaye et al., in press) [33]
Line 114: phosphoric acid is H3PO4
Thanks for catching this oversight – it is corrected
Line 183: Rimmer et al. [45] paper is only peripherally related to the question of PH3 in Venusian clouds or the debate if the Greaves et al signal is PH3 or SO2. While Rimmer et al results are very intriguing they do not help directly in settling the debate if the Greaves et al signal is PH3 or SO2. I would rephrase it a little bit to make that clear.
The wording has been adjusted
Line 199: Mogul’s reanalysis, if correct, potentially suggests much more complex disequilibria than just those that arise from PH3 and H2S. It might be worth pointing that out. As atmospheric disequilibria are commonly cited as a possible biosignature.
Thank you. Good point. It is done.
Line 207: of course, such hypothetical metabolism must then regenerate NADH
Agreed
Line 223-225: discusses results of a retracted arXiv paper. The authors of the retracted paper left the comment that justified the retraction: “The authors discovered that a preprint on ArXiV with a submission date prior to ours contains, buried deep within it, the same idea.”
The arXiv submission that “contained the same idea” as presented in Turong and Lunine is in fact Bains, W.; Petkowski, J.J.; Seager, S.; Ranjan, S.; Sousa-Silva, C.; Rimer, P.B.; Zhan, Z.; Greaves, J..; Richards, A.M.S. Phosphine on Venus cannot be explained by conventional processes. Arxiv 2020, https://arxiv.org/abs/2009.06499
Bains et al 2020 address the ideas of Turong and Lunine in their paper and conclude that it is an implausible pathway to the formation of PH3. I would advise against citing of the retracted papers as this might confuse the reader and add the false credibility to the retracted papers. Otherwise, there should be a note that “the paper was since retracted as the ideas presented in it were addressed originally in Bains et al 2020.”
Thank you, I wasn´t aware of it. That reference has been removed
Line 278: The Dallol geothermal area is nowhere near the scidities of Venusian clouds. This should be emphasized more clearly. Citations for the Dallol should also be added. The author may also consider adding discussion, and citation, of the Iron Mountain in California (another hyperacidic environment – in Earth terms of that phrase of course).
The wording makes this clear as is, but a reference (Gomez at al. 2019) has been added. Adding the Iron Mountain Site or other sites would deviate from the focus and the point made, thus didn´t further elaborate.
Line 285-287: The subsequent re-analysis of Greaves (by the same authors) suggest potential patchy distribution of PH3.
Yes, certainly heterogeneous (that´s why global average and peak concentrations are given), but difficult to say due to being so close to the detection limit. Line 301 states now the heterogenous distribution.
Line 317-319: the formation of P2H4 could help in distinguishing between abiotic and biotic sources of PH3 (P2H4 is formed in lab conditions simulating lightning together with PH3) also if phosphine is made by hydrolysis of mineral phosphides, then we would also expect diphosphine (P2H4) to be generated. – the citations for possible association of P2H4 with life is needed, as far as the reviewer knows it is the other way around – the presence of P2H4 signifies some abiotic component reaction (it does not mean automatically that PH3 is non-biological of course).
The source is difficult to delineate. After further looking into it, I agree, that it is not clear whether the P2H4 would be a biotic or abiotic source. So, I changed the sense to point out that photolysis of PH3 must be a significant process in the Venusian atmosphere (with Ferris and Benson reference provided). If PH3 is at significant concentrations, no matter what the source is, it would be useful to also detect it in Venusian spectra to verify PH3.
Line 333: I would say temperate cloud decks instead of the lower Venusian atmosphere, as not all of the lower atmosphere is suitable for life.
Yes, good idea. Done.
Other points:
Milojevic, T.; Treiman, A.; Limaye, S. Phsphorus in Venus clouds. In review at Astrobiology. 438 – there is a typo in Phosphorus.
Typo is corrected
Thank you so much for the thorough review !

Reviewer 2 Report
Peer Review – MDPI Life, Section Astrobiology
Manuscript ID: 1155139
Manuscript Title: The Case (or not) for Life in the Venusian Clouds
Manuscript Author: Prof. Dr. Dirk Schulze-Makuch
Major Comments
This manuscript is an excellent idea delivered with eloquence. I like the framework the author has used, somewhere between a comment and a review. He has summarised what has been contributed to the scientific literature on this topic since September 2020, in addition to the authors own insights based upon his interest in this field for the past two decades. Furthermore, I think the author does a good job of guiding us through the background literature on the conceptual issues surrounding Venusian habitability, then onto the active discussion of the phosphine discovery itself. I have only a few comments/suggestions to make below, and then a number of minor corrections that I’d like to see made. Overall, I thoroughly endorse this manuscript for publication in Life.
- The included figure only provides limited additional insight compared to the main text, the only real value being that it highlights the presence of the unknown UV absorbers. I believe this article could be enhanced by some form of summary figure that visually collates the information presented by the author in the manuscript. My suggestion would be to include a figure that summarises the known habitability constraints of the Venusian atmosphere.
- I would include an additional reference (see below), but overall I believe the author has done an admirable job collating the literature published in the last 7 months. Perhaps include the reference below at line 253?
Cockell, C. S., McMahon, S., & Biddle, J. F. (2020). When Is Life a Viable Hypothesis? The Case of Venusian Phosphine. Astrobiology, 21(3), 261-264.
- Perhaps highlight some potential experimental/observational pathways towards understanding how organisms could adapt to the extreme conditions of the Venusian middle cloud deck. Searching for new extreme sulfuric acid environments on Earth? Experimental evolution of known extremophiles (e.g. acidophiles) to increasingly extreme sulfuric acidic conditions? These are just my suggestions, but I’d be interested to hear yours.
Minor Comments
L21 - Change to: “on Venus”, or “in the Venusian atmosphere”.
L32 – This sentence is vague. Specify the exact environment analysed (aerial water droplets?) and which organisms were studied. Bacterial cell division?
L33 – Remove gap.
L57 - Change to: “as a potential”.
L67 – Provide specific name for Pioneer probe (I appreciate there are numerous names, choose one) e.g. Pioneer-Venus.
L77 – Remove the first “of”.
L85 – Change to: “mastering them”.
L87 – Change to: “maybe”.
L88 – Change to: “within Venusian aerosols”.
L112 – Change to: “not only found sulfur, but”.
L132 – A square bracket is missing, please rectify this.
L189 – A full stop is missing at the end of the sentence.
L197-202 – I would clarify here that Mogul et al. state they have found evidence for PH3 and/or H2S, as opposed to PH3 and H2S. This is an important distinction, given the controversy surrounding the Greaves et al. findings.
L242 – Can equation 2 be fit onto one line?
L284 – Change the “it” to “they” in this sentence, if you are referring to multiple pathways, which is how the sentence is set up.
L306 – Change to “solutions”.
L322 – Include Cockell 1999 in this reference list.
Line 324 – Re-phrase, perhaps to: “deliquescence as their sole source of water”.

Author Response
Response to Reviewer 2
Peer Review – MDPI Life, Section Astrobiology
Manuscript ID: 1155139
Manuscript Title: The Case (or not) for Life in the Venusian Clouds
Manuscript Author: Prof. Dr. Dirk Schulze-Makuch
Major Comments
This manuscript is an excellent idea delivered with eloquence. I like the framework the author has used, somewhere between a comment and a review. He has summarised what has been contributed to the scientific literature on this topic since September 2020, in addition to the authors own insights based upon his interest in this field for the past two decades. Furthermore, I think the author does a good job of guiding us through the background literature on the conceptual issues surrounding Venusian habitability, then onto the active discussion of the phosphine discovery itself. I have only a few comments/suggestions to make below, and then a number of minor corrections that I’d like to see made. Overall, I thoroughly endorse this manuscript for publication in Life.
Thank you very much
- The included figure only provides limited additional insight compared to the main text, the only real value being that it highlights the presence of the unknown UV absorbers. I believe this article could be enhanced by some form of summary figure that visually collates the information presented by the author in the manuscript. My suggestion would be to include a figure that summarises the known habitability constraints of the Venusian atmosphere.
Thank you so much for the suggestion. Very helpful. A summary figure with these parameters is now included. I hope that is what the reviewer had in mind
- I would include an additional reference (see below), but overall I believe the author has done an admirable job collating the literature published in the last 7 months. Perhaps include the reference below at line 253?
Cockell, C. S., McMahon, S., & Biddle, J. F. (2020). When Is Life a Viable Hypothesis? The Case of Venusian Phosphine. Astrobiology, 21(3), 261-264.
The reference is now included and discussed. That paper is more a philosophical standpoint and I decided before not to get into that kind of discussion (how far should authors go with their claims), but think now that it is actually useful to discuss and thus included it
Perhaps highlight some potential experimental/observational pathways towards understanding how organisms could adapt to the extreme conditions of the Venusian middle cloud deck. Searching for new extreme sulfuric acid environments on Earth? Experimental evolution of known extremophiles (e.g. acidophiles) to increasingly extreme sulfuric acidic conditions? These are just my suggestions, but I’d be interested to hear yours.
My suggestions are now included (line 353 to line 369). Again, thank you for the comment
Minor Comments
L21 - Change to: “on Venus”, or “in the Venusian atmosphere”. done
L32 – This sentence is vague. Specify the exact environment analysed (aerial water droplets?) and which organisms were studied. Bacterial cell division? It is now specified
L33 – Remove gap. done
L57 - Change to: “as a potential”. done
L67 – Provide specific name for Pioneer probe (I appreciate there are numerous names, choose one) e.g. Pioneer-Venus. Done
L77 – Remove the first “of”. done
L85 – Change to: “mastering them”. done
L87 – Change to: “maybe”. This part has been reworded
L88 – Change to: “within Venusian aerosols”. No, actually it is “at” according to the reference cited
L112 – Change to: “not only found sulfur, but”. done
L132 – A square bracket is missing, please rectify this. done
L189 – A full stop is missing at the end of the sentence. done
L197-202 – I would clarify here that Mogul et al. state they have found evidence for PH3 and/or H2S, as opposed to PH3 and H2S. This is an important distinction, given the controversy surrounding the Greaves et al. findings.
I checked with Dr. Mogul on this and it is actually “and”, the “and/or” was in an earlier version of the manuscript
L242 – Can equation 2 be fit onto one line? done
L284 – Change the “it” to “they” in this sentence, if you are referring to multiple pathways, which is how the sentence is set up.
L306 – Change to “solutions”. done
L322 – Include Cockell 1999 in this reference list. He is referenced earlier and not here, because he did not propose in his paper some kind of mechanism how life adapts
Line 324 – Re-phrase, perhaps to: “deliquescence as their sole source of water”. done
Thank you so much for the constructive and helpful review !
Reviewer 3 Report
The communication paper entitled The Case (or not) for Life in the Venusian Clouds by Dirk Schulze-Makuch discusses the chemical conditions in the Venusian atmosphere, as measured in the past with the recent findings of significant phosphine concentrations, under the angle of the potential presence of life in the lower layers of the Venusian atmosphere. The author raises caution as on the limitations we face, the lack of data and repeatability. He also explores all environmental conditions in the Venusian clouds and whether they could sustain life. Taking previous knowledge for earth-like life, argues that the latter could sustained, however the detection of phosphine alone (if measurements are correct) as a biosignature is not enough evidence to argue in favour of a potential presence of life in the lower layers of the Venusian atmosphere, highlighting the luck of detection of other by-products of the (potentially biotic) chemical reaction that results into the phosphine synthesis. This well-written manuscript is an important contribution to the discussion on whether life could adapt and survive on the Venusian Clouds. I would only ask the author to elaborate and comment on the combination of all the different metabolic capabilities this super-bug would have in order to be able to survive. Apart from that, no plagiarism was detected using an anti-plagiarism software, and I suggest it is accepted after correcting certain minor typos, as in line 39 (area to era) and line 189 (full stop missing).
Author Response
Response to Reviewer 3
The communication paper entitled The Case (or not) for Life in the Venusian Clouds by Dirk Schulze-Makuch discusses the chemical conditions in the Venusian atmosphere, as measured in the past with the recent findings of significant phosphine concentrations, under the angle of the potential presence of life in the lower layers of the Venusian atmosphere. The author raises caution as on the limitations we face, the lack of data and repeatability. He also explores all environmental conditions in the Venusian clouds and whether they could sustain life. Taking previous knowledge for earth-like life, argues that the latter could sustained, however the detection of phosphine alone (if measurements are correct) as a biosignature is not enough evidence to argue in favour of a potential presence of life in the lower layers of the Venusian atmosphere, highlighting the luck of detection of other by-products of the (potentially biotic) chemical reaction that results into the phosphine synthesis. This well-written manuscript is an important contribution to the discussion on whether life could adapt and survive on the Venusian Clouds.
Thank you
I would only ask the author to elaborate and comment on the combination of all the different metabolic capabilities this super-bug would have in order to be able to survive.
Not sure what the reviewer means. All major possible metabolic pathways or all adaptations that a major metabolic pathway would have to have? Either way, given our lack of knowledge of the specific atmospheric conditions at Venus, we simply cannot say at this point. What I did instead to improve clarity was to expand Fig. 1 to show the major positive habitability parameters of the Venusian clouds and the major negative ones, plus the scientific speculations how some of the negative ones, the challenges, can be overcome. This is where we are standing right now regarding our knowledge.
Apart from that, no plagiarism was detected using an anti-plagiarism software, and I suggest it is accepted after correcting certain minor typos, as in line 39 (area to era) and line 189 (full stop missing).
The typos have been corrected – thank you
Reviewer 4 Report
See attached file

Author Response
Response to Reviewer 4
General point of view: thank you
1.Water activity:
this has been explained and elaborated as suggested by the reviewer, see lines 75 to 84.
2.Venus and the habitable zone:
Actually, the statement is true. Our star is a G dwarf which increases in luminosity with time. Thus, Venus used to be in the habitable zone and now Earth is in it, and in the future, in a few billion years, Earth won´t be anymore in the habitable zone, and we will have a similar fate as Venus. Only M dwarf stars have a relatively stable habitable zone. But the reviewer is absolutely correct that being in the defined habitable zone does not mean that the planet is actually habitable. A half sentence has been included on line 56 to make this a bit more clear.
- Clear wording in sentences:
Thank you, that is very helpful. All the suggestions were implemented except
Line 17, my dictionary says unhabitable is an old, obsolete term, so the term used was not changed
Line 18, this is meant to be conditional case so “would have” is correct
Line 51, imported would imply human action, so transported is more neutral
Line 85, not changed because it would change the intended meaning
Line 93, implemented, but used “Earth” rather “terrestrial”, because terrestrial implies land (Terran, I think, would then be the correct term, which, however, sounds SciFi)
- Contractions
Have been removed
- Chemical Formulas for P compounds
These have been added
- Interior of aerosols
From spacecraft measurements we know that the aerosols are heterogeneous, and it is pointed out in the paper that the interior is coated with some sulfur material. Thus, it would be important to know what the interior is composed of – dust or perhaps organic compounds?
- Commas
I addressed this. However, I´m not a native speaker, so would think that any remaining mistakes will be taken care of by the journal before publication
- Other minor comments
Thank you, all was corrected, except
Line 115-116, C,N,P because these elements were discussed in the cited paper (Cockell et al. 1999)
Line 244, the “with” is correct
Line 247, yes, but what is meant is that there is another follow-up reaction that can occur after the equation (3) reaction
Thank you for your detailed review, which was extremely helpful !
Reviewer 5 Report
The manuscript is well written and informative. However, it would do better with some minor changes to improve its readability.
- There should be some graphic illustrations, namely having one to identify Venus's atmosphere layers; and one showing the transition of Venus from the habitable zone to its present placing, with some elaborate captions.
- In lines 254-256, this sentence (or something similar) should be on top of the introduction. Perhaps adding some years would be helpful too.
- While it's only natural that most of "life" referred to in this manuscript is based on life-as-we-know-it, perhaps the author should give a strong caveat early on suggesting the limitation of having n=1. One or two lines in the introduction and at the discussion should warrant this.
Author Response
Response to Reviewer 5
The manuscript is well written and informative. However, it would do better with some minor changes to improve its readability.
Thank you
- There should be some graphic illustrations, namely having one to identify Venus's atmosphere layers; and one showing the transition of Venus from the habitable zone to its present placing, with some elaborate captions.
The temperate cloud layer is thought to be at an elevation of 48 to 60 km above the surface, but we don´t know for sure about the exact elevations. How the transitions happened on Venus from habitable conditions to today is unknown, mostly because of the near-global resurfacing that happened about 700 million years ago on Venus (as pointed out in the paper)
What I did instead was expand Fig. 1 to show the major positive habitability parameters of the Venusian clouds and the major negative ones, plus the scientific speculations how some of the negative ones, the challenges, can be overcome.
- In lines 254-256, this sentence (or something similar) should be on top of the introduction. Perhaps adding some years would be helpful too.
A similar wording is right at the beginning of the second paragraph of the introduction to set the tone after the entry paragraph, but it also makes sense to talk about it in the given context
- While it's only natural that most of "life" referred to in this manuscript is based on life-as-we-know-it, perhaps the author should give a strong caveat early on suggesting the limitation of having n=1. One or two lines in the introduction and at the discussion should warrant this.
A paragraph has been added to address this point (line 365 to line 369)